# Analysis of Pharmacy Cardiac Optimization Clinic for Patients with New Onset Atrial Fibrillation Detected via Cardiac Implantable Electronic Device Clinic

**DOI:** 10.3390/pharmacy11020048

**Published:** 2023-03-03

**Authors:** Ellen Schellhase, Madeline Stanko, Natalie Kinstler, Monica L. Miller, Sotiris Antoniou, Sadeer Fhadil, Mital Patel, Paul Wright

**Affiliations:** 1Purdue University College of Pharmacy, West Lafayette, IN 47907, USA; 2St. Bartholomew’s Hospital, London EC1A 7BE, UK

**Keywords:** anticoagulation, atrial fibrillation, cardiac implantable electronic devices, cardiac pharmacists, medication optimization, direct oral anticoagulants, non-vitamin k oral anticoagulants, cardiovascular optimization

## Abstract

For patients with cardiac implantable electronic devices (CIEDs), arrythmias such as atrial fibrillation (AF) can be detected and actions taken to rapidly assess and initiate treatment where appropriate. Actions include timely initiation of anticoagulation, review of blood pressure, and optimization of cholesterol/lipids to prevent unfavorable outcomes, such as stroke and other cardiovascular complications. Delays to initiating anticoagulation can have devastating consequences. We sought to implement a virtual clinic, where a pharmacist reviews patient referrals from a CIED clinic after detecting AF from the CIED. Anticoagulation choice is determined by patient-specific factors, and a shared patient–provider decision to start oral anticoagulation is made. In addition, blood pressure readings and medications are assessed with lipid-lowering therapies for optimization. A total of 315 patients have been admitted through this clinic and anticoagulated over a two-year span; in addition, 322 successful interventions were made for optimization of cardiac therapy. Rapid initiation of anticoagulation within five days of referral was likely to have reduced unfavorable outcomes, such as stroke and other cardiovascular optimizations, leading to improved patient outcomes.

## 1. Introduction

Stroke is a devastating consequence of atrial fibrillation (AF) [1]. Oral anticoagulation has been shown to reduce the risk of stroke by 64% and the risk of death by 26% in patients with AF [2,3]. Both American Heart Association/American College of Cardiology (AHA/ACC) and the European Society of Cardiology (ESC) Guidelines recommend initiation of oral anticoagulant (OAC) therapy in patients with AF with a CHA_2_DS_2_VASc score ≥2 (males) and ≥3 (females) [3,4], with consideration for initiating when scores are ≥1 (males) and ≥2 (females), respectively. Moreover, bleeding risk scores such as HAS-BLED and ORBIT are used to assess risk of bleeding and to identify reversible risk factors to support safer initiation of OAC therapy [5,6]. A HAS-BLED score of 0 indicates low risk, 1–2 indicates moderate risk, and ≥3 indicates high risk of major bleeding) [7]. While this scoring system is well validated, these scores are only a generalized representation of risk and do not include all risk factors that can contribute to stroke/bleeding. Therefore, it is important that patients are individually assessed for ischemic and bleeding risks prior to initiation of appropriate anticoagulation at appropriate doses to maximize therapeutic outcome [8,9].

Many patients with AF or who have short runs of AF are asymptomatic and, as such, diagnosis can be difficult [10]. Often, asymptomatic patients are noted to have AF during serendipitous screening, such as ECGs taken prior to surgical procedures or from screening programs for selected high-risk individuals. For patients with cardiac implantable electronic devices (CIED)—including permanent pacemakers (PPM), implantable cardiac defibrillators (ICD), and cardiac resynchronization therapy (CRT) devices—in addition to their primary function, these devices record continuous electrical activity of the heart [11]. When patients have their device “downloaded”, arrythmias such as AF can be detected and actions to assess and treat can be undertaken [4].

Most CIEDs are able to complete a “remote” download, which means patients are not required to complete an in-person visit to the cardiac center or clinic. At St. Bartholomew’s Hospital, a tertiary referral hospital, the CIED clinic serves a large geographical area, covering northeast London and other external referrals. Typically, around 95% of patients complete remote downloads in the comfort of their own home, which can decrease financial costs for both patients and healthcare systems while improving patient satisfaction [12]. Previously, once AF was detected and a decision was made to anticoagulate, patients’ general practitioners (GP) were informed, via a clinical letter, of the diagnosis and guidance to have the patient start anticoagulation. GPs would then refer patients to their local hospital anticoagulation clinic for medication initiation. This process provided several opportunities for slow- or no-care initiation due to ongoing delays between these steps, or patients potentially being lost to follow-up [13,14,15,16,17].

To address these challenges, a pharmacy cardiac optimization clinic (PCOC) was developed at St. Bartholomew’s Hospital. This is a pharmacist-led medication optimization clinic aimed at further reducing adverse cardiovascular outcomes and related hospitalizations [18]. This is a new service and care model, for which this paper aims to share both the clinic model and outcomes for enrolled patients.

## 2. Materials and Methods

This was a retrospective-chart review of all patients enrolled in the PCOC. Data collection included: demographics, date of CIED detection of arrhythmia, date of PCOC appointment, medication start date, current medications, medications that were adjusted, CHA_2_DS_2_VASc score, HAS-BLED score, ORBIT score, and QRISK. All PCOC patient charts were reviewed and included in this report. Descriptive statistics were used to evaluate data.

### 2.1. PCOC Structure

When a patient’s CIED was interrogated, a report was generated and reviewed by a device technician. When AF was detected, a medical doctor further reviewed the report along with the patient medical record to determine if the patient should be referred to the PCOC for initiation of anticoagulation. The average time to referral through initiation of the PCOC was 5 days, and referrals were assessed daily by a specialized cardiac pharmacist. The referral pathway is further described in Figure 1.

PCOC inclusion criteria included: all patients with CIED-detected AF/flutter, and a CHA_2_DS_2_-VASc score of equal to or greater than 1 for males and equal to or greater than 2 for females.

The PCOC was set up just before the first wave of the COVID-19 pandemic. At this time, there was an increase in remote monitoring and, as such, remote detection of AF [19]. The PCOC was adapted to the need to provide telehealth and had become exclusively a telephone-based service. The initial telephone call with a patient usually took between 30–40 min to establish the risks and benefits of anticoagulation therapy and to identify the most appropriate anticoagulant. Additionally, requests for bloods (if no recent results available) were organized.

The PCOC service was undertaken as part of daily duties within the cardiac pharmacy team, with one pharmacist tasked with reviewing email referrals and making initial contact with new patients. As part of the monitoring, patients were contacted again after 4–6 weeks to assess medication adherence and to check for adverse effects, such as nuisance bleeding. If all was well, an onward transfer of care to their local GP was organized.

### 2.2. PCOC Responsibilities

During a virtual appointment, the pharmacist completed the following activities:A cardiovascular review of patient-specific factors. Such factors included: sex, age, weight, smoking status, indication, comorbidities, left ventricular ejection fraction, previous incidence of myocardial infarction or stroke, blood pressure, HbA1c, full blood count, lipid panel, renal function, liver function, international normalized ratio, CHA_2_DS_2_VASc score, HAS-BLED score, ORBIT score, QRISK, assess need for concomitant, and all current medications [20,21,22].Initiation of anticoagulation. The choice of anticoagulant was facilitated through discussion with the patient and in consideration of individual patient needs. Almost exclusively, a direct oral anticoagulant (DOAC) was selected based on several patient-related factors.Assessment of lipid therapy. If there was established cardiovascular disease, the pharmacist would ensure that a high-intensity statin was prescribed (namely atorvastatin 80 mg daily). If there was no established CVD, the 10-year cardiovascular risk (using QRISK) was assessed, and if the risk was greater than 10% then initiation of atorvastatin 20 mg daily was considered, in addition to lifestyle advice [23,24].Review of anti-hypertensive therapy. Titration was considered where clinically indicated [25,26].

## 3. Results

A total of 315 patients were referred, met the criteria for the PCOC, and were anticoagulated between October 2020 and August 2022. Table 1 shows the baseline demographics of the patients referred to the PCOC. The mean CHA_2_DS_2_VASc for females was 4.2 and the mean score for males was 3.5. The mean HAS-BLED score was 1.2. The ORBIT score was recorded starting in November 2021 in response to the updated NICE guidance for AF diagnosis and management, which recommends the ORBIT bleeding score as the tool of choice to assess bleeding risk [27]. The average ORBIT score suggested that 69.3% of referrals were low bleed risk, 10.1% were medium bleed risk, and the remaining 20.6% were high bleed risk.

Following patient consultation, all 315 patients were initiated on anticoagulation, with 194 placed on edoxaban, 101 on apixaban, 19 on rivaroxaban, and one on dabigatran (Figure 2).

There were 322 successful interventions made for optimization of cardiac therapy. Forty-one (13%) patients were initiated on a statin, and 46 (14.6%) patients already established on a statin had their doses optimized. Changes to rate control were made in 20 (6.3%) patients. Twenty-six (8.3%) patients were either started on antihypertensives or had their doses optimized. Upon referral, 190 (60.3%) patients were on antiplatelet therapy and 97.3% (n = 185) were stopped after pharmacist review.

## 4. Discussion

During the PCOC initiation and start-up, initial referral rates averaged five per month, while the average time to anticoagulation start was approximately three days [13]. Currently, the PCOC consistently averages 15 referrals each month and the average time to anticoagulation is five days. Although this is still vastly more optimal than the patients’ anticoagulation timelines before the start of the clinic, the increase in average time to anticoagulation is thought to be due to increasing patient volume while the PCOC staffing level has remained the same. The virtual clinic model, which was initially implemented in response to the need for remote care during the COVID-19 pandemic, has continued. The convenience for the patient may also contribute to the overall improvement in the time to anticoagulation compared with the traditional referral through the GP.

The patient demographics highlight that the mean CHA_2_DS_2_VAS_c_ scores were consistently well above the threshold for initiating anticoagulation. Additionally, the mean HASBLED score has remained consistent since the clinic started. It is encouraging to see that the appropriate patients are being referred to the PCOC for anticoagulation initiation and optimization of cardiac medications. When we reviewed patients with high ORBIT scores, 63% of these patients were taking an antiplatelet treatment that was reviewed and stopped prior to anticoagulation initiation, hence minimizing the bleeding risk.

The majority of patients tolerated anticoagulation initiation, with only a small number requiring further intervention at the follow-up appointment. There were 44 reports of adverse events to medication changes from the PCOC, with 22 related to anticoagulation. From the 22 related to anticoagulation, only two patients discontinued therapy due to bleeding issues and 13 successfully switched to an alternative agent (7 from apixaban and 6 from edoxaban). The remainder of reported adverse events were gastrointestinal upset, rash, and headache, which did not require stoppage of therapy.

An NHS England publication-commissioning recommendation for national procurement for DOACs, released at the start of 2022, advises that edoxaban should be used where clinically appropriate, and to consider other DOACs if there are contraindications to edoxaban or other clinical indications [28]. Prior to this recommendation update, use of any DOAC was preferred over warfarin, taking into consideration bleeding risk, contraindications and tolerance [27]. The anticoagulant choice had no significant impact in terms of adverse effects and adherence to the medication.

It should be noted that one specialized cardiac pharmacist was directly responsible for PCOC duties, which were undertaken prior to usual clinical care responsibilities within the hospital wards. While other specialized clinical pharmacists helped to operate the clinic when needed, there was room for expansion of personnel within the clinic. Increased personnel involvement could help decrease the time to anticoagulation back to three days or fewer, and provide more timely responses to email referrals and dataset updates. The pharmacists who contributed to this service found it professionally rewarding, as they were able to utilize their clinical knowledge and engage in collaborative practice.

## 5. Conclusions

The PCOC has created a new and innovative opportunity for cardiac specialty pharmacists to engage with patients, collaborate with GPs, decrease time to anticoagulation, and provide cardiac medication optimization. The results to date demonstrated the high impact of pharmacist-led outpatient follow-up clinics. This clinical service provided a model that can be used by other health systems when implementing CIED-detected AF/anticoagulation management and cardiac medication optimization protocols. The PCOC has demonstrated an impact on patient care by decreasing time to anticoagulation which may reduce the occurrence of unfavorable outcomes, including incidence of stroke, as well as further reducing health consequences by optimization of cardiac medications. This is the future of health-system pharmacy, where the pharmacist’s independent clinical responsibility is an integral part of cardiovascular care and risk reduction in an outpatient setting.

## Figures and Tables

**Figure 1 pharmacy-11-00048-f001:**
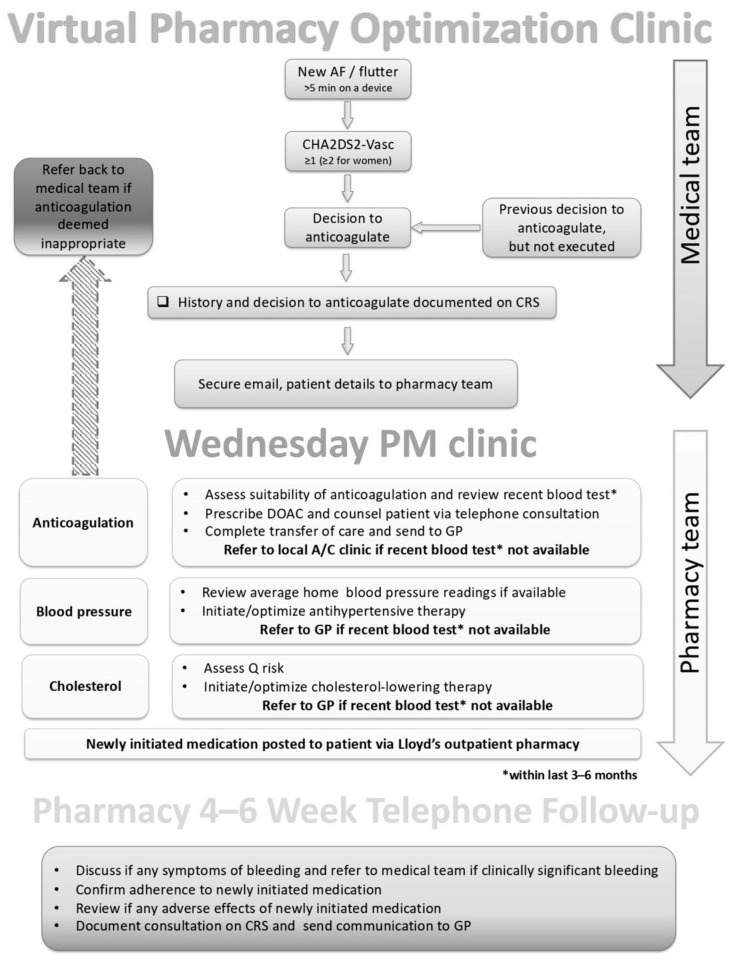
Virtual Pharmacy Cardiac Optimization Clinic pathway.

**Figure 2 pharmacy-11-00048-f002:**
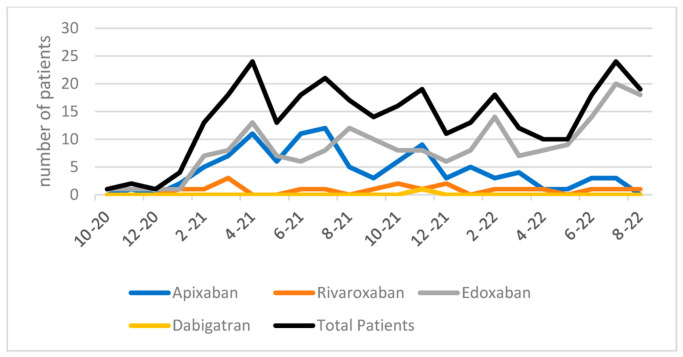
Anticoagulation initiated following PCOC enrollment.

**Table 1 pharmacy-11-00048-t001:** Baseline demographics.

Mean Age		76 years old (42–100 years)
Sex	Male	225 (71.4%)
	Female	90 (28.6%)
Current Smokers		26 (8.3%)
Mean CHA_2_DS_2_VAS_c_	Male	3.5
	Female	4.2
	% with LVSD/HCM	101 (32.0%)
	% with hypertension	209 (66.3%)
	% with diabetes	87 (27.6%)
	% with history of CVA/TIA	51 (16.1%)
	% with history of IHD/PVD	124 (39.4%)
Mean HASBLED		1.2
On lipid therapy at time of PCOC referral		234 (74.3%)
On antiplatelet therapy at time of PCOC referral		190 (60.3%)

PCOC, Pharmacy Cardiac Optimization Clinic; LVSD/HCM, left ventricular systolic dysfunction/hypertrophic cardiomyopathy; CVA/TIA, cerebrovascular accident/transient ischemic attack; IHD/PVD, ischemic heart disease/peripheral vascular disease.

## Data Availability

Not applicable.

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
