# Peer review of "Analysis of Pharmacy Cardiac Optimization Clinic for Patients with New Onset Atrial Fibrillation Detected via Cardiac Implantable Electronic Device Clinic"

_pharmacy, 2023, doi:10.3390/pharmacy11020048_

Round 1

Reviewer 1 Report

The Authors focused on the study of Analysis of Pharmacy Cardiac Optimization Clinic for Patients with New Onset Atrial Fibrillation Detected via Cardiac Implantable Electronic Device Clinic. I consider that the idea of this communication is very interesting and with important clinical and pharmaceutical implications.

This is generally a well-written and comprehensive communication, with the clear figures, interesting findings that can make significant contributions to further large studies.

In my opinion:

- The abstract presents an accurate description of this study.

- The Authors was conducted adequate literature review.

- References support the rationale for reporting the study.

- Patients are described adequately.

- The management of the study is effectively described.

- Valid and reliable outcome measures were used.

- Conclusions are appropriate and comprehensive.

- >2 or > 2 - please decide on one version and standardize throughout the publication.

Author Response

See attached file with comment-by-comment responses. 

Reviewer 2 Report

I would like to thank all the authors for their idea to implement a virtual clinic where pharmacists can review patient referrals from a device clinic after detecting AF from the CIED. The concept is excellent; however, I have significant concerns about this work:

- First, I think this work could fit in public health journals as it focuses more on policy change. Although tried by pharmacists but it requires approval by everyone involved in the medical community to make it a general policy that can be tried everywhere.

- Second, As initially, those patients have cardiac implantable electronic devices, which means they are at high risk and need proper assessment by their cardiologists before starting medications; the authors didn't comment on requesting echocardiography before starting anticoagulation which will be helpful to know if any heart pathology behind the AF or it is just standalone AF.

- One of my major concerns is that the authors claimed that this study did not require ethical approval. I think IRB approval is a must here as the study is based on patient data and requires the patient's consent to join the clinic and accept treatment options from the pharmacist (not his primary physician).

Although the clinic was very helpful in cutting the referral time short, especially during the COVID pandemic however, the authors didn't mention in the manuscript, especially the discussion of any new methods that were established for AF detection, such as Apple Watch or other smart devices that can be used to alert the physician or pharmacist through AI algorithm in the clinic so they can review patient records and could be applied to the general population, not just CIED patients.

Lastly, minor comments: very long introduction, no need for lipid therapy assessment as it is not urgent in this situation and can be reviewed by PCP after liver function reassessment, especially with the anticoagulation.

Author Response

(The authors gave the same response as above.)

Reviewer 3 Report

The authors investigate AF diagnosis and therapy initiation thanks to CID. This retrospective analyses shows how there is a need for monitoring of patients for onset of AF in a population at high risk. The study is overall interesting however major points must be addressed. 

Population description is missing! What were the CVDs of these patients? 

Concomitant therapy is missing!

Author Response

(The authors gave the same response as above.)

Round 2

Reviewer 2 Report

The authors responded to my initial concerns. Thanks.

Reviewer 3 Report

The authors have made the required adjustments. The article is now fit for publication.